# Synergistic Activity of Cefiderocol in Combination with Avibactam, Sulbactam or Tazobactam against Carbapenem-Resistant Gram-Negative Bacteria

**DOI:** 10.3390/cells13161315

**Published:** 2024-08-06

**Authors:** Russell E. Lewis, Marta Palombo, Erica Diani, Benedetta Secci, Davide Gibellini, Paolo Gaibani

**Affiliations:** 1Department of Molecular Medicine, University of Padova, 35122 Padova, Italy; russelledward.lewis@unipd.it; 2IRCCS Azienda Ospedaliero, Universitaria di Bologna, 40135 Bologna, Italy; marta.palombo@aosp.bo.it (M.P.);; 3Department of Diagnostic and Public Health, Microbiology Section, Verona University, 37134 Verona, Italy; erica.diani@univr.it (E.D.); davide.gibellini@univr.it (D.G.)

**Keywords:** carbapenem-resistant *Enterobacteriales*, carbapenem-resistant *Pseudomonas aeruginosa*, carbapenem-resistant *Acinetobacter baumannii*, cefiderocol/avibactam, cefiderocol/sulbactam, cefiderocol/tazobactam

## Abstract

We investigated the activity of cefiderocol/β-lactamase inhibitor combinations against clinical strains with different susceptibility profiles to cefiderocol to explore the potentiality of antibiotic combinations as a strategy to contain the major public health problem of multidrug-resistant (MDR) pathogens. Specifically, we evaluated the synergistic activity of cefiderocol with avibactam, sulbactam, or tazobactam on three of the most “Critical Priority” group of MDR bacteria (carbapenem-resistant *Enterobacterales*, *Pseudomonas aeruginosa*, and *Acinetobacter baumannii*). Clinical isolates were genomically characterized by Illumina iSeq 100. The synergy test was conducted with time-kill curve assays. Specifically, cefiderocol/avibactam, /sulbactam, or /tazobactam combinations were analyzed. Synergism was assigned if bacterial grow reduction reached 2 log_10_ CFU/mL. We reported the high antimicrobial activity of the cefiderocol/sulbactam combination against carbapenem-resistant *Enterobacterales*, *P. aeruginosa*, and *A. baumannii*; of the cefiderocol/avibactam combination against carbapenem-resistant *Enterobacterales*; and of the cefiderocol/tazobactam combination against carbapenem-resistant *Enterobacterales* and *P. aeruginosa.* Our results demonstrate that all β-lactamase inhibitors (BLIs) tested are able to enhance cefiderocol antimicrobial activity, also against cefiderocol-resistant isolates. The cefiderocol/sulbactam combination emerges as the most promising combination, proving to highly enhance cefiderocol activity in all the analyzed carbapenem-resistant Gram-negative isolates, whereas the Cefiderocol/tazobactam combination resulted in being active only against carbapenem-resistant *Enterobacterales* and *P. aeruginosa,* and cefiderocol/avibactam was only active against carbapenem-resistant *Enterobacterales.*

## 1. Introduction

Difficult-to-treat (DTR) infections due to multidrug-resistant (MDR) Gram-negative bacteria are serious threats worldwide and represent a challenge for clinicians due to reduced treatment options available for patients with multiple co-morbidities [1]. The emergence of resistance to carbapenem reduces the antimicrobial armamentarium available to treat infection sustained by MDR pathogens. In 2017, the World Health Organization (WHO) defined *Mycobacterium tuberculosis*, *Salmonella*, *Shigella*, *Neisseria gonorrhoeae*, *Staphylococcus aureus, Pseudomonas aeruginosa*, *Acinetobacter baumannii*, and carbapenem-resistant (CR) *Enterobacterales* (CRE) among the most “Critical Priority” group of MDR bacteria, for which new antibiotics are urgently needed [2]. In the last few years, new β-lactam/β-lactamase inhibitor (βL-βLIC) combinations have been developed to treat infections sustained by CR Gram-negative bacteria [3]. Nevertheless, the emergence of strains resistant to novel antibiotic combinations has been described lately, resulting in a reduction in the therapeutic options available [4]. FDA and EMA have recently authorized the clinical use of a novel cephalosporin, cefiderocol (CFD) [5,6], to treat urinary tract infections (UTIs), bacterial hospital-acquired pneumonia (HAP), and ventilator-associated pneumonia (VAP) [5]. CFD demonstrated high in vitro antimicrobial activity against *Klebsiella pneumoniae*, *Escherichia coli*, *P. aeruginosa*, and *A. baumannii* [7], showing intrinsic structural stability toward β-lactamases, such as *K. pneumoniae* carbapenemase (KPC) [8], oxacillin carbapenemase (OXA), New Delhi metallo-β-lactamase (NDM), and Verona integron metallo-β-lactamase (VIM) [9]. Distinctively to other cephalosporin, CFD is a siderophore compound able to cross the outer membrane of Gram-negative bacteria through the siderophore iron uptake system [6]. This peculiarity confers to CFD protection against mutations, as efflux pumps’ overexpression and porins’ downregulation or truncation, able to alter outer membrane permeability [6]. However, the emergence of CFD-resistant strains has recently been reported. In particular, cross-resistance between ceftazidime/avibactam and CFD has been observed, combined with KPC variants in *K. pneumoniae* and also in NDM-harboring strains [10,11,12]. In this context, drug combinations may be considered a valuable approach to improve treatment options and hinder the emergence of CR strains. The characterization of novel antibiotic combinations based on CFD added to β-lactamase inhibitors (βLI) may represent an attractive strategy able to associate the “Trojan horse” entry strategy into bacterial cells of CFD together with the inhibition of β-lactamases activity [13]. Among βLI, avibactam, usually matched with ceftazidime, is effective against class A (extended spectrum β-lactamase [ESBL], KPC), class C (AmpC), and class D (OXA-48-like) β-lactamases [14]. First-generation βLI, sulbactam, is a penicillanic acid sulfone, showing its activity mostly against class A β-lactamases and penicillinases, plasmid-mediated ESBL, and chromosomal AmpC (12). Sulbactam has lately been combined with cefoperazone, which is more effective against Gram-negative bacteria infections, especially against *P. aeruginosa* and *A. baumannii* [15], while sulbactam combined with durlobactam could represent an encouraging antibiotic for patients with MDR *A. baumannii* infections [16]. The cephalosporin ceftolozane, acting by binding to penicillin-binding protein (PBP) and inhibiting bacterial membrane synthesis, was recently combined with the classic βLI tazobactam as a new antimicrobial combination against *P. aeruginosa*. However, the resistance of *P. aeruginosa* to this combination of compounds seems to be related to the modification of the AmpC enzyme structure and to PBP3 mutations [17]. Herein, we evaluate the in vitro activity of new therapeutic combinations of CFD with three clinical βLIs (avibactam, sulbactam, and tazobactam) against MDR clinical isolate, specifically CR *K. pneumoniae*, CR *P. aeruginosa* (CR-Pa), and CR *A. baumannii* (CR-Ab).

## 2. Materials and Methods

Strains included in this study were extensively described in our previous study [13]. Bacterial strains’ identification was performed by the MALDI-TOF MS assay (Bruker Daltonics, Bremen, Germany). Antimicrobial susceptibility testing was performed using the MicroScan Walkaway system. The minimum inhibitory concentration (MIC) was confirmed by Sensititre^TM^ Plate EUMDRXXF (Thermofisher, Waltham, MA, USA) and microdilution in iron-depleted Mueller–Hinton broth (ID-MHB) with the ComASP Cefiderocol Test (Liofilchem, Roseto degli Abruzzi, Italy), following the European Committee on Antimicrobial Susceptibility Testing (EUCAST) clinical breakpoints v13.0 (available at https://www.eucast.org/clinical_breakpoints/, accessed on 2 January 2023). Carbapenemase production was identified by the NG-Test CARBA 5 (NG Biotech, ZI Courbouton, Guipry-Messac, France) and confirmed with molecular assay Xpert Carba-R (Cepheid, Sunnyvale, CA, USA).

### 2.1. Genomic Analysis

Genomic DNA was extracted from purified bacterial cultures of *K. pneumoniae*, *P. aeruginosa*, and *A. baumannii* using the DNeasy Blood&Tissue Kit (Qiagen, Hombrechtikon, Switzerland) and cleaned up with AMPure XP magnetic beads (Beckman Coulter, Brea, CA, USA). Bacterial genome libraries were prepared with the DNA Prep Library Preparation Kit (Illumina, San Diego, CA, USA) and sequenced by the Illumina iSeq 100 platform (Illumina, San Diego, CA, USA) using iSeq Reagent Kit v.2 with 2 × 150 paired-end reads. The read sets’ quality was evaluated using the FastQC v12.1 software (https://www.bioinformatics.babraham.ac.uk/projects/fastqc/, accessed on 1 January 2023). Genomes were assembled with SPAdes v.3.10 and polished with Pilon v.1.23. Antimicrobial resistance genes and MLST analysis were evaluated using an online platform (available at: http://genomicepidemiology.org/services/, accessed on 1 January 2023). Genome assembly and annotation statistics were performed by custom Python scripts based on the Biopython v1.79 package (https://biopython.org/, accessed on 1 January 2023). Genes involved in antimicrobial resistance detection were obtained with AMRFinderPlus v3.10.30 (https://github.com/ncbi/amr, accessed on 1 January 2023). Porins genes were investigated by aligning nucleotide sequences against the respective amino acid sequence retrieved from the NCBI database (https://github.com/bbuchfink/diamond, accessed on 1 January 2023). The analyzed sequences were as follows: OmpK35 [WP_004141771.1] and OmpK36 [WP_002913005.1] for *K. pneumoniae*; OprD [WP_000910004] and CarO [ABC46545.1] for *A. baumannii*; and OprD [P32722.1] and OprF [P13794.1] for *P. aeruginosa*.

### 2.2. Time-Kill Curve Assay

Time-kill assays were performed by inoculating 5 × 10^5^ CFU/mL of organism in iron-depleted cation-adjusted Mueller–Hinton broth. The CFD was tested alone and in combination with avibactam, sulbactam, and tazobactam. For each strain, two concentrations of CFD were evaluated, equal to the MIC and ½ the MIC. Avibactam was added to the bacterial culture at a final concentration of 1, 4, and 16 mg/L, whereas sulbactam and tazobactam were tested at final concentrations of 2, 4, 8, and 16 mg/L. To determine the viable bacteria count during the time-kill experiments, 10 µL of each suspension was removed, serially diluted 10-fold in PBS, and plated on blood agar plates at defined time points (0, 2, 4, 8, and 24 h post-inoculation). The bacterial growth (log_10_ CFU/mL) was determined by enumerating the colonies after 18–22 h of the plates’ incubation at 37 °C. The lower limit of accurately quantifiable CFU using this method was 2 log_10_ of viable bacteria per mL. Each experiment was performed in triplicate for the principal time points (i.e., 0, 4, and 24 h) and singularly for the middle time points.

### 2.3. Synergy Analysis

Interactions of CFD with βLI were analyzed using surface response methods for drug synergy analysis as implemented in the BIGL package for R version 4.01 (R Core Team (2020). R: A language and environment for statistical computing. R Foundation for Statistical Computing, Vienna, Austria. URL, last access: 1 January 2024; available at: https://www.R-project.org/). Unlike traditional analysis (e.g., chequerboard, FICI index), the Loewe model as implemented in the BIGL package can account for drugs with different maximal antimicrobial activities (as expected with CFD and βLI) during the evaluation of pharmacodynamic interactions [18]. Synergy analysis was performed in three steps. First, consultation–response curves were constructed for CFD and βLI tested alone by fitting the logarithmically transformed time-kill curve assay data from the 24 h time point using a 4-parameter logistic model constrained to the same baseline response to determine the marginal dose response curves for CFD and βLI independently at test concentrations spanning 0–512 mg/L. Parameters were estimated using a non-linear least squares regression approach. Marginal parameters were subsequently used to construct a generalized Loewe model (null response) surface. Finally, the expected null response was compared to observed 24 h CFU/mL counts for each antibiotic concentration/combination tested in time-kill curve analysis by the bootstrapped analysis of each analysis point in relation to the null model to define synergistic (greater than expected killing) or antagonistic (less than expected killing) activity [15]. Synergy analysis was summarized for each CFD/βLI by plotting the reported Log_10_ CFU/mL difference from the null response with a 95% CI as a composite heatmap.

## 3. Results

### 3.1. Bacterial Characterization

In this study, we selected six clinical strains with different susceptible profiles to CFD. A total of (n = 2) *K. pneumoniae* CRE, (n = 2) CR-Pa, and (n = 2) CR-Ab were selected based on the susceptibility profile to CFD (i.e., susceptible and resistant). The CR isolates’ phenotypic traits, resulting in AST, are reported in Appendix A. The genomic characterization of the strains included in this study is summarized in Table 1. 

### 3.2. Synergy Results with Time-Kill Curve (TKC) Assay

The synergy tests results, performed with the time-kill curve (TKC) assay against CRE, CR-Pa, and CR-Ab, are reported in Figure 1. 

CFD and avibactam showed synergism when tested against CRE, whereas no synergic interaction was reported against CR-Pa and CR-Ab strains. Detailed results obtained with CRE showed a strong reduction in bacterial growth in both the CFD-susceptible and -resistant isolates when avibactam was added to CFD equal to the MIC. The log_10_ fold reduction in CFU/mL was of 4.41, 6.22, and 6.22 for the CFD-susceptible strain at avibactam 1, 4, and 16 mg/L and 6.30 for the CFD-resistant isolate at all the avibactam concentrations tested. The same analysis performed at a CFD concentration equal to ½ the MIC highlighted a different trend between the CFD-susceptible and -resistant strain. The bacterial count fold reduction was >2 (4.27, 5.54, and 6.70) log_10_ CFU/mL with all concentrations of avibactam in the CFD-resistant isolate, whereas a significant reduction (2.48 log_10_ CFU/mL) was reached in the CFD-susceptible strain only with 16 mg/L of avibactam. When analyzed by time-kill curves, sulbactam strongly enhanced CFD antibacterial activity against CRE, CR-Pa, and CR-Ab. A total of 2 mg/L of sulbactam, added to the CFD concentration equal to the MIC, was sufficient to obtain a reduction in bacterial count greater than 2 log_10_ CFU/mL in all the isolates analyzed. The CFU/mL log_10_ fold reduction was 5.70 and 6.30 in the CFD-susceptible and -resistant CRE, 6.70 and 6.40 in the CFD-susceptible and -resistant CR-Pa, and 3.40 and 6.70 in the CFD-susceptible and -resistant CR-Ab. At concentrations of CFD equal to ½ the MIC and sulbactam of 2 mg/L, CFD-resistant CRE and CR-Pa, and both CFD-susceptible and -resistant CR-Ab, had a significant bacterial count reduction (>2 log_10_ CFU/mL). The fold reduction recorded was 6.70, 5.20, 6.70, and 4.10 log_10_ CFU/mL, respectively. No reduction in bacterial count was reported at CFD equal to ½ the MIC in the CFD-susceptible CRE and CR-Pa. Tazobactam substantially improved CFD antimicrobial activity against CRE and CR-Pa, whereas a small enhancement of CFD activity was detected against CR-Ab. In CRE, the fold reduction collected with 2 mg/L of tazobactam and CFD equal to the MIC was 5.54 and 6.00 log_10_ CFU/mL in the CFD-susceptible and -resistant strains, whereas it reached 6.10 log_10_ CFU/mL when CFD was equal to ½ the MIC in the CFD-resistant isolate. No reduction in growth was observed in the CFD-susceptible isolate with CFD equal to ½ the MIC. In CR-Pa, 2 mg/L of tazobactam strongly decreased the bacterial count with CFD equal to the MIC in the CFD-susceptible isolate and with CFD equal to the MIC and ½ the MIC in the CFD-resistant isolate. The fold reduction recorded was 5.00, 6.00, and 5.90 log_10_ CFU/mL, respectively. A total of 8 mg/L of tazobactam added to CFD equal to ½ the MIC significantly reduced bacterial growth in the CFD-susceptible strain, with a 3.52 log_10_ CFU/mL fold reduction. The CFD/tazobactam combination, tested against CR-Ab, significantly affected bacterial growth in the CFD-susceptible isolate when CFD was equal to the MIC and tazobactam was 2 mg/L (2.28 log_10_ CFU/mL). No considerable bacterial count decrease was reported at a CFD concentration equal to ½ the MIC. Against CFD-resistant CR-Ab, 16 mg/L of tazobactam was required to significantly inhibit bacterial growth at CFD equal to the MIC and ½ the MIC (2.02 and 2.09 log_10_ CFU/mL, respectively). 

### 3.3. Synergy Analysis

Combinations analysis using the generalized Loewe null response model confirmed synergistic interactions of antimicrobial killing when βLI concentrations of 1–16 mg/L were combined with CFD (Figure 2). 

The greatest negative mean R deviation from the null response model (synergistic interactions) was evident when CFD was combined with avibactam against CRE and sulbactam against CFD-susceptible and CFD-resistant strains of *K. pneumoniae*, *A. baumanii*, and *P. aeruginosa*. Tazobactam exhibited synergistic effects against both CFD-susceptible and -resistant CRE and *P. aeruginosa*, with weaker synergistic effects against *A. baumannii*. 

## 4. Discussion

Herein we reported the in vitro antimicrobial activity of CFD in combination with avibactam, sulbactam, or tazobactam against CRE, CR-Pa, and CR-Ab performed with the TKC assay. Overall, our results showed the synergic antimicrobial activity of the CFD/sulbactam combination against CRE, CR-Pa, and CR-Ab. The addition of sulbactam proved to highly enhance CFD activity in all the analyzed CR Gram-negative strains, both CFD-susceptible and -resistant ones. Our findings demonstrated strong synergic interaction between CFD and avibactam when tested against CRE isolates. The CFD/tazobactam combination showed high antimicrobial activity against CRE and CR-Pa and low activity against CR-Ab. Based on these results, we can hypothesize that these combinations could be active against CR-Pa strains showing emerging resistance to cefiderocol in vivo [19]. In agreement with these results, previous studies reported against MDR strains of *A. baumannii* marked CFD antimicrobial activity in combination with ampicillin/sulbactam both in vitro and in vivo using human-simulated regimens in the murine infection model [20] and synergism between CFD and avibactam, sulbactam, or tazobactam in vitro [21]. In particular, in our previous study, we demonstrated that in vitro synergic interaction between CFD and ceftazidime/avibactam was reported against CRE and CR-Pa and between CFD and ampicillin/sulbactam among CR-Pa [13]. At the same time, we observed that CFD in association with tazobactam or avibactam exerted increasing antibacterial activity against CR-Pa, CR-Ab, and CRE by different methods [22], thus showing comparable results [13]. Limited clinical data have also indicated that the combined administration of CFD, ampicillin/sulbactam, and tigecycline resulted in a beneficial effect against infection sustained by extensively drug-resistant (XDR)/CR *A. baumannii*, and an in vitro test performed against the same isolate showed synergism among CFD, sulbactam, and tigecycline [22,23]. The in vitro synergic interaction between CFD and avibactam or tazobactam was formerly described against KPC-producing *Enterobacterales*, predominantly *K. pneumoniae* [22]. 

The present study is one of the first investigations of the in vitro antibacterial activity of CFD- and βLI-based antibiotic combinations against three of the most “WHO Critical Priority” group of MDR bacteria (CRE, CR-Pa, and CR-Ab). This study has the same limitations. The number of strains analyzed is small, and the number of βLI tested should be increased. Thus, additional studies would be advantageous to obtain a comprehensive characterization of the bactericidal activity of the CFD/βLI combinations against MDR Gram-negative pathogens. Also, we observed a regrowth of strains between 8 h and 24 h for different antimicrobial combinations and for CFD alone (Figure 1). It is possible to hypothesize that a resistant subpopulation could emerge under sublethal concentrations of cefiderocol that could be related to emerging resistant clones. Further studies should be carried out to evaluate the presence of resistant subpopulations against different combinations. Lastly, we cannot suggest the use of TKC in a classical routine workflow principally due to the laborious and time-consuming work of this method. Therefore, we could hypothesize that the E-test method could be used for synergy testing in a bacteriological laboratory during routine workflow [13]. However, E-test method has different limitations that should be considered during the interpretation of the results. Our advice is that further studies should be carried out to implement the TKC assay to include this method in a bacteriological routine workflow. 

## 5. Conclusions

In conclusion, our data demonstrated the ability of all βLIs analyzed (avibactam, sulbactam, and tazobactam) to enhance CFD antimicrobial activity against CRE. A comparable synergic interaction was observed when avibactam, sulbactam, or tazobactam were added to CFD and tested against CRE. The TKC assay performed against CR-Pa revealed high synergistic antimicrobial activity with both the CFD/sulbactam and CFD/tazobactam combinations. Our data indicated that the CFD/sulbactam combination was the most active against CR-Ab, although less synergic interaction was reported between CFD and tazobactam.

## Figures and Tables

**Figure 1 cells-13-01315-f001:**
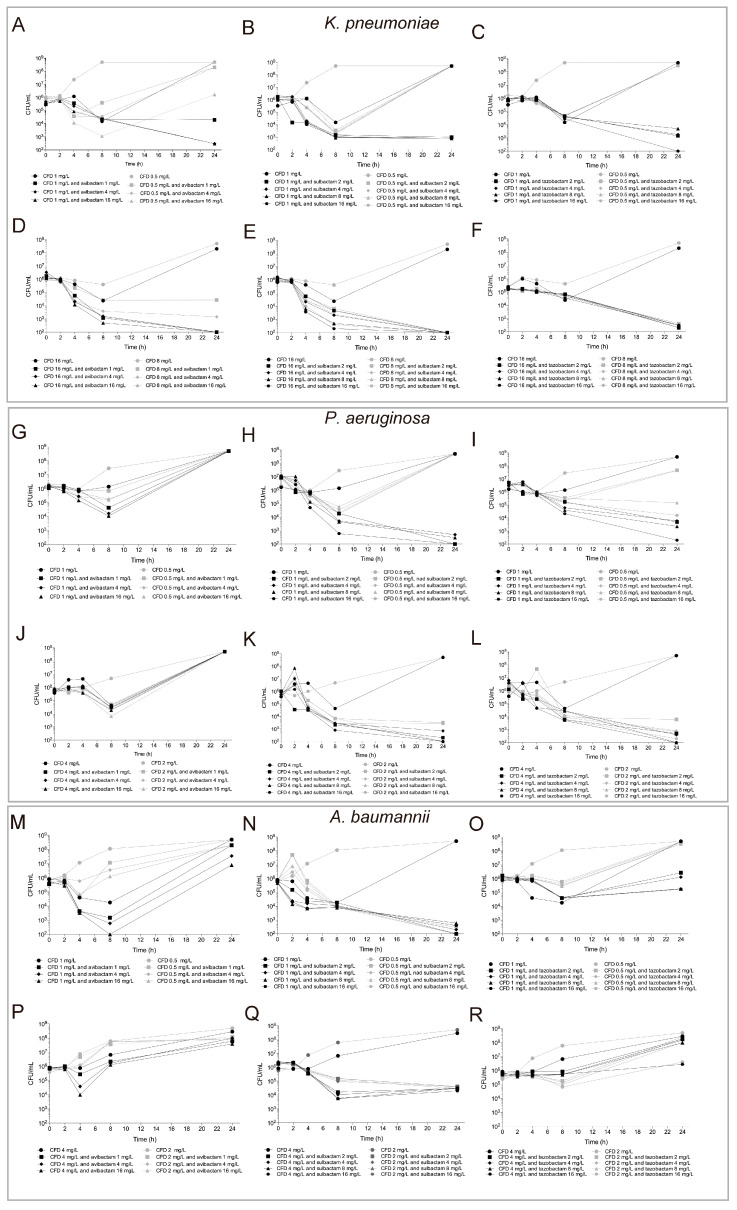
TKC obtained for CRE, CFD-susceptible (**A**–**C**), and CFD-resistant (**D**–**F**); for CR-Pa, CFD-susceptible (**G**–**I**), and CFD-resistant (**J**–**L**); and for CR-Ab, CFD-susceptible (**M**–**O**), and CFD-resistant (**P**–**R**) with CFD in combination with avibactam (**A**,**D**,**G**,**J**,**M**,**P**), sulbactam (**B**,**E**,**H**,**K**,**N**,**Q**), and tazobactam (**C**,**F**,**I**,**L**,**O**,**R**).

**Figure 2 cells-13-01315-f002:**
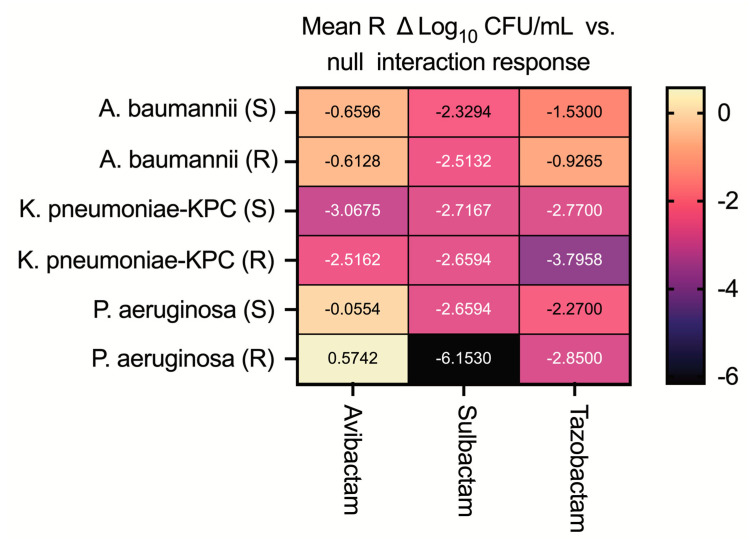
The average (mean R) CFU/mL deviation from null response interaction model for tested cefiderocol (CFD)-β-lactamase inhibitor combinations at 24 h determined by time-kill curve analysis. Negative mean R values are indicative of average synergistic effects; positive values are indicative of average antagonistic effects.

**Table 1 cells-13-01315-t001:** Genotypic characteristics of CR strains included in this study.

Major Porins	Multidrug Efflux Pumps	β-Lactamase	ST	Isolate
OmpK35, truncated at aa 229; OmpK36, truncated at aa 183	*emrD*, *oqxA*, *oqxB1*	KPC-3, TEM-1, SHV-28,	307	CRE CFD-susceptible
OmpK35, truncated at aa 41; OmpK36, INS135GD	*emrD*, *oqxA*, *oqxA*	KPC-66, TEM-1, SHV-11, OXA-10, OXA-181, CMY-16	512	CRE CFD-resistant
oprD, variant C2, truncated at aa 64, G425A; oprF, wt	*mexA*, *mexE*, *mexX*	ADC-73, TEM-1, OXA-23, OXA-66, ftsl	2	CR-Pa CFD-susceptible
oprD, variant C2, G425A; oprF, wt	*mexA*, *mexE*, *mexX*	ADC-73, OXA-23, OXA-66	2	CR-Pa CFD-resistant
carO, variant III, Y245F; oprD, wt	*adeC*, *amvA*	OXA-2, OXA-488, PDC-35, PER-1	235	CR-Ab CFD-susceptible
carO, variant III, Y245F; oprD, wt	*adeC*, *amvA*	OXA-848, BEL, PDC-16	298	CR-Ab CFD-resistant

## Data Availability

Data is available upon request.

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
