# Peer review of "Synergistic Activity of Cefiderocol in Combination with Avibactam, Sulbactam or Tazobactam against Carbapenem-Resistant Gram-Negative Bacteria"

_cells, 2024, doi:10.3390/cells13161315_

Round 1

Reviewer 1 Report

Comments and Suggestions for Authors

"Synergistic activity of Cefiderocol in combination with Avibactam, Sulbactam or Tazobactam against Carbapenem-Resistant Gram-Negative Bacteria," 

The study describes the synergistic effects of combining cefiderocol with β-lactamase inhibitors-avibactam, sulbactam, and tazobactam-against carbapenem-resistant Klebsiella pneuminiae, Pseudomonas aeruginosa, and Acinetobacter baumannii. The research found that these combinations enhance cefiderocol’s activity, particularly cefiderocol/sulbactam.

The study discusses the enhanced antimicrobial activity of cefiderocol when combined with β-lactamase inhibitors, particularly sulbactam. It aligns the findings with previous studies and emphasizes the potential clinical implications in treating MDR infections.

The study is well-structured, with a clear research question, robust methodology, and comprehensive results. However, the limitation is the small sample size of bacterial strains with various STs.  Additionally, the study focuses solely on in vitro assays, and in vivo validations would further substantiate the findings.

Minor:

line 27. against

Fig. 1. too small to see the numerics and symbols. enlarge the figures and increase resolution

Author Response

The study describes the synergistic effects of combining cefiderocol with β-lactamase inhibitors-avibactam, sulbactam, and tazobactam-against carbapenem-resistant Klebsiella pneuminiae, Pseudomonas aeruginosa, and Acinetobacter baumannii. The research found that these combinations enhance cefiderocol’s activity, particularly cefiderocol/sulbactam.

The study discusses the enhanced antimicrobial activity of cefiderocol when combined with β-lactamase inhibitors, particularly sulbactam. It aligns the findings with previous studies and emphasizes the potential clinical implications in treating MDR infections.

The study is well-structured, with a clear research question, robust methodology, and comprehensive results. However, the limitation is the small sample size of bacterial strains with various STs.  Additionally, the study focuses solely on in vitro assays, and in vivo validations would further substantiate the findings.

Authors’ reply and amendments: We thank the reviewer for his work and for his comments. We grouped the clinical strains on the basis of the antimicrobial profile for the resistance to cefiderocol, thus selecting a resistant and a susceptible isolates. Considering the low resistance rate to cefiderocol among CR-Ab and CR-Pa it was difficult to obtain more strains. At the same time, most of the CRE strains collected in our institution were clonally related, thus reducing the sample size. Also, further specific studies are ongoing to validate clinically the results presented here

ine 27. against

Authors’ reply and amendments:  The term was added as requested

Fig. 1. too small to see the numerics and symbols. enlarge the figures and increase resolution

Authors’ reply and amendments: The figure is uploaded separately to improve the resolution and to clear shown the symbols

Reviewer 2 Report

Comments and Suggestions for Authors

Dear Authors,
I have read your article titled "Synergistic Activity of Cefiderocol in Combination with Avibactam, Sulbactam or Tazobactam against Carbapenem-Resistant Gram-Negative Bacteria" with great interest.
While reading, I noticed that the name of the bacterium Acinetobacter baumannii is not consistently spelled correctly. In some parts of the text (for example, Figure 1 or page 7, line 207), it appears as "Acinetobacter baumanii". Since accurate nomenclature is essential for scientific precision and to avoid confusion, I wanted to bring this typographical error to your attention.
I kindly request you to review the article and correct the instances where the spelling is incorrect.
I am confident that addressing this detail will further enhance the quality of your already excellent work.

Author Response

Authors’ reply and amendments: We thank the reviewer for his comments. The nomenclature and any typos were revised throughout the entire manuscript, as requested

Reviewer 3 Report

Comments and Suggestions for Authors

This study shows the activity of some beta-lactamase inhibitors (sulbactam, tazobactam, avibactam) associated with cefiderocol against six MDR Gram negative bacteria from Italy. Time-kill studies was performed to study these drug associations.

First of all, it seems that the authors conducted a synergy study in 2022-2023 on the same six strains using the combined Etest method (Antibiotics (Basel). 2023 May; 12(5): 858., ref 18 from the present study): the WGS data from Table 1 of the previous study are absolutely identical (ST, betalactamases, efflux pumps, porin mutations) to the WGS data reported in Table 1 here. Therefore I conclude that these were the exactly same strains?

-  This reference should be mentioned in the introduction

- Synergy tests from the previous study (using combined Etests) should be thoroughly compared to TK experiments results in the discussion part

- There is a point I do not understand: In the previous study, FDC MICs are 0.125 and 0.5 for both PA strains, respectively; in the present study, MICs are 1 and 4mg/L, respectively.

Other comments:

-TK studies are known to be difficult to reproduce. Were these experiments performed at least in duplicate? If several assays were performed, error bars should appear on the graphs.

-line 51. Some studies highlighted the higher resistance rates observed with NDM-harboring strains versus other carbapenemases (KPC, VIM, OXA) (see e.g. Antimicrob Agents Chemother. 2020 Nov 17;64(12):e01582-20.doi: 10.1128/AAC.01582-20., or Antimicrob Agents Chemother. 2016 Jul; 60(7): 4384–4386.). This data could be added to the introduction.

-line 80. Since few strains were tested, some common resistance mechanisms found in MDR bacteria are not represented here. For example, authors insist on the carbapenemase issue (VIM, NDM, OXA, KPC) in their introduction, and some carbapenemase classes (NDM, VIM) are not even represented here. Morover, both the PA strains do not carry any clinical significant carbapenemases (OXA-848 and -488 are OXA-50 derivatives, which is the chromosomal oxacillinase from PA which generally does not confer any high level resistance to betalactams). What were selection criteria for these six strains? Were they randomly selected? For example, considering MDR PA strains from Italy, I would have expected at least one metalloenzyme (e.g. VIM)-carrying PA.

- Table 1. It seems that results for PA strains and ABA strains are inverted, please correct the Table.

-Line 148. I do not manage to access Supp data. Anyway, some AST results should be included for each strain, at least for main betalactam classes, which are the main focus of the study (e.g. ampicillin, ampicillin+sulbactam, piperacillin, piperacillin-tazobactam, ceftazidime, ceftazidime+avibactam, aztreonam, cefepime, imipenem, ceftolozane+tazobactam, cefiderocol).

-Fig 1. I do not know if it is due to my computer, but the quality of Fig 1 is very poor and difficult to read. I would suggest to enlarge each graph (maybe put two graphs horizontally instead of three) and/or  upload this Figure with better quality.

-Fig1. When there is synergy, it is all the more present at 24 rather than at 8h. It seems that there is a “regrowth” of strains between 8h and 24h when using FDC alone. Such a curve aspect could evoke the emergence of FDC-resistant mutants. Was it explored by the authors? Did they at least perform AST on strains isolated at 24h to see if these had acquired additional resistance to FDC? Maybe the association of betalactamase inhibitor hampers the emergence of high-level resistance to FDC?

-line 229-230. When you look at ref 20, the association of FDC+AVI or TZB was also evaluated against P. aeruginosa and A. baumannii, it could be added to the discussion and compared to results from the present study.

- Despite the few strains tested, these drug combinations seem promising. However, time-kill studies remain research methods which are difficult to implement routinely in most microbiology labs. Do the authors perform these tests in routine? What method would they suggest for routine in vitro testing of such associations against clinical isolates?

Minor comments

-line 18. Sulbactan=>sulbactam

-line 21. baumanni=>baumannii

-Fig 1. baumanii=>baumannii

-There are several typo mistakes throughout the abstract, please re-read carefully.

-line 71-73. Please rephrase. The binding to PBP and inhibition of membrane synthesis is due to ceftolozane itself, not its association with tazobactam.

-Table 1. CRE FDC-R: What is TEM number? What means “oqx” alone, since you also show “oqxA” ?

Comments on the Quality of English Language

There are some minor typo errors to address in the abstract

Author Response

First of all, it seems that the authors conducted a synergy study in 2022-2023 on the same six strains using the combined Etest method (Antibiotics (Basel). 2023 May; 12(5): 858., ref 18 from the present study): the WGS data from Table 1 of the previous study are absolutely identical (ST, betalactamases, efflux pumps, porin mutations) to the WGS data reported in Table 1 here. Therefore I conclude that these were the exactly same strains?

Authors’ reply and amendments: We thank the reviewer for his comments. Most of the strains used in this manuscript were characterized in our previous study. However, in this manuscript we tested different antimicrobial combinations and also using reference and more accurate testing method for synergy (i.e. TKC).

- This reference should be mentioned in the introduction

Authors’ reply and amendments: The reference was added to the introduction section as requested

- Synergy tests from the previous study (using combined Etests) should be thoroughly compared to TK experiments results in the discussion part 

Authors’ reply and amendments: A sentence better describing our previous results was added to the discussion section as requested.

- There is a point I do not understand: In the previous study, FDC MICs are 0.125 and 0.5 for both PA strains, respectively; in the present study, MICs are 1 and 4mg/L, respectively. 

Authors’ reply and amendments: The discrepancies observed by the reviewer between the MICs in these studies were due to the different testing methods used. In particular, in previous study we tested all the strains by using by e-test method, while in this study we performed the microdilution method using Iron Depleted-Mueller Hinton Broth (ID-MHB)

Other comments:

-TK studies are known to be difficult to reproduce. Were these experiments performed at least in duplicate? If several assays were performed, error bars should appear on the graphs.

Authors’ reply and amendments: The TKC experiments for antimicrobial combinations were performed in triplicate for the principal time points (0, 4 and 24 hours) to confirm the data and in single for the middle time points. Thus, we opted to maintain the figure in this format without SD or SE bars in order to simplify the readers and for clarity.

-line 51. Some studies highlighted the higher resistance rates observed with NDM-harboring strains versus other carbapenemases (KPC, VIM, OXA) (see e.g. Antimicrob Agents Chemother. 2020 Nov 17;64(12):e01582-20.doi: 10.1128/AAC.01582-20., or Antimicrob Agents Chemother. 2016 Jul; 60(7): 4384–4386.). This data could be added to the introduction.

Authors’ reply and amendments: a sentence describing previous results and the suggested references were added to the text as requested

-line 80. Since few strains were tested, some common resistance mechanisms found in MDR bacteria are not represented here. For example, authors insist on the carbapenemase issue (VIM, NDM, OXA, KPC) in their introduction, and some carbapenemase classes (NDM, VIM) are not even represented here. Morover, both the PA strains do not carry any clinical significant carbapenemases (OXA-848 and -488 are OXA-50 derivatives, which is the chromosomal oxacillinase from PA which generally does not confer any high level resistance to betalactams). What were selection criteria for these six strains? Were they randomly selected? For example, considering MDR PA strains from Italy, I would have expected at least one metalloenzyme (e.g. VIM)-carrying PA. 

Authors’ reply and amendments: We thanks reviewer for his insightful comment. We opted to use clinical strains collected in our institution by selecting a couple of susceptible and resistant strains for each species. Indeed, we collected few resistant strains among CR-Pa and CR-Ab and most of them were clonally related by showing identical resistome profiles. Thus we opted to focalize to the couple of susceptible and resistant clinical strains and focalizing the study to these novel combinations by testing with TKC method

- Table 1. It seems that results for PA strains and ABA strains are inverted, please correct the Table. 

Authors’ reply and amendments: We apologize for the mistake. We modify the Table 1 accordingly to the correct antimicrobial resistance genes for each species

-Line 148. I do not manage to access Supp data. Anyway, some AST results should be included for each strain, at least for main betalactam classes, which are the main focus of the study (e.g. ampicillin, ampicillin+sulbactam, piperacillin, piperacillin-tazobactam, ceftazidime, ceftazidime+avibactam, aztreonam, cefepime, imipenem, ceftolozane+tazobactam, cefiderocol). 

Authors’ reply and amendments: We apologize for the mistake. We uploaded the supplementary material with comparators (16 antimicrobial molecules tested) as requested

-Fig 1. I do not know if it is due to my computer, but the quality of Fig 1 is very poor and difficult to read. I would suggest to enlarge each graph (maybe put two graphs horizontally instead of three) and/or  upload this Figure with better quality.

Authors’ reply and amendments: The figure is uploaded separately to improve the resolution and to better shown the single figures

-Fig1. When there is synergy, it is all the more present at 24 rather than at 8h. It seems that there is a “regrowth” of strains between 8h and 24h when using FDC alone. Such a curve aspect could evoke the emergence of FDC-resistant mutants. Was it explored by the authors? Did they at least perform AST on strains isolated at 24h to see if these had acquired additional resistance to FDC? Maybe the association of betalactamase inhibitor hampers the emergence of high-level resistance to FDC?

Authors’ reply and amendments: We thanks reviewer for his insightful comment. We tested the strains at 24h for cefiderocol alone and we didn’t observed any substantial difference rather than parental strains. At the same time, it is possible to hypothesize that resistant subpopulation could emerge under sublethal concentrations of cefiderocol that could related to emerging resistant clones. However, we didn’t tested this hypothesis and further study will be conducted to better evaluate this important point raised by the reviewer

-line 229-230. When you look at ref 20, the association of FDC+AVI or TZB was also evaluated against P. aeruginosa and A. baumannii, it could be added to the discussion and compared to results from the present study. 

Authors’ reply and amendments: A sentence describing our previous results with cefiderocol in combination with tazobactam or avibactam was added to the discussion section n order to compare the two study.

- Despite the few strains tested, these drug combinations seem promising. However, time-kill studies remain research methods which are difficult to implement routinely in most microbiology labs. Do the authors perform these tests in routine? What method would they suggest for routine in vitro testing of such associations against clinical isolates?

Authors’ reply and amendments: We agree with the reviewer comment, that the TKC cannot be used in a classical routine workflow principally based on the laborious and time-consuming work for this type of experiment. Based on our experience, we could hypothesize that e-test method could be use for synergy testing in a bacteriological laboratory during the routine workflow. However, E-test method have different limitations that should be considered during the results interpretations. In our advise, further studies should be done to implement TKC assay to include this method in a bacteriological routine workflow.

Minor comments

-line 18. Sulbactan=>sulbactam

Authors’ reply and amendments: The term was modified as suggested

-line 21. baumanni=>baumannii

Authors’ reply and amendments: The term was modified as suggest

-Fig 1. baumanii=>baumannii

Authors’ reply and amendments: The term was modified as suggest

-There are several typo mistakes throughout the abstract, please re-read carefully.

Authors’ reply and amendments: The abstract was revised to modify all the typo

-line 71-73. Please rephrase. The binding to PBP and inhibition of membrane synthesis is due to ceftolozane itself, not its association with tazobactam. 

Authors’ reply and amendments: the sentence was modified accordingly to the reviewer suggestion

-Table 1. CRE FDC-R: What is TEM number? What means “oqx” alone, since you also show “oqxA” ?

Authors’ reply and amendments: We apologize for the mistake. We modify the antimicrobial resistance determinants in the Table 1 accordingly to the gene variants.

Comments on the Quality of English Language

There are some minor typo errors to address in the abstract 

Authors’ reply and amendments: the manuscript was entirely revised to modify any typos

Reviewer 4 Report

Comments and Suggestions for Authors

In the manuscript ID: cells-3071082 the authors investigate the synergistic effect of the associations of cefiderocol with the β-lactamase inhibitors avibactam, sulbactam and tazobactam. The reported assays, performed on cefiderocol-susceptible/-resistant, carbapenem-resistant strains of Klebsiella pneumoniae, Pseudomonas aeruginosa and Acinetobacter baumannii, evidenced a higher efficacy of the combination cefiderocol/sulbactam, which potentiated the drug bactericidal activity against all the tested isolates. The authors conclude that such combinations could represent promising options even to counteract the emerging resistance to cefiderocol.

The manuscript is well conceived and structured, with a comprehensive introduction, a detailed method section, reporting updated techniques and a good result presentation. The discussion supports the described data and properly places them in the body of the existing literature, which is still very recent regarding cefiderocol resistance. There are just few minor corrections that should be performed to further improve the manuscript quality:

·        Please check the correspondence between the main reported efflux pumps and the related bacterial species in Table 1, especially for P. aeruginosa and A. baumannii, as they should be inverted;

·        Please provide a better-quality Figure 1, even evidencing the most significant bacterial load reductions;

·        Please comment in the discussion the synergistic activity of β-lactamase inhibitors with cefiderocol considering the proposed resistance mechanisms that could be counteracted by the inhibitors themselves (doi: 10.1007/s10096-022-04526-0 );

·        Please make the supplementary material  available;

·        A minor revision of the English form is recommended.

Once performed these corrections, the paper can be accepted for publication.

MINOR COMMENTS

·        Line 23, please specify the abbreviation “BLI” or use the specific terms in the abstract;

·        Line 127, please type “e.g.” in italic;

·        Line 152, please explain the “TKC” abbreviation.

Comments on the Quality of English Language

Minor errors in the English form to be checked and corrected.

Author Response

·Please check the correspondence between the main reported efflux pumps and the related bacterial species in Table 1, especially for P. aeruginosa and A. baumannii, as they should be inverted;

Authors’ reply and amendments: We thank the reviewer for his comments. Table 1 was revised in order to correct the typos.

·Please provide a better-quality Figure 1, even evidencing the most significant bacterial load reductions;

Authors’ reply and amendments: The figure is uploaded separately to improve the resolution and to clear shown the symbols

-Please comment in the discussion the synergistic activity of β-lactamase inhibitors with cefiderocol considering the proposed resistance mechanisms that could be counteracted by the inhibitors themselves (doi: 10.1007/s10096-022-04526-0 );

Authors’ reply and amendments: A sentence describing the proposed synergistic interactions against resistant strains was added to the discussion section as requested

·Please make the supplementary material  available;

Authors’ reply and amendments: we apologize for the mistake. We uploaded the supplementary material as requested

·A minor revision of the English form is recommended.

Authors’ reply and amendments: The text was entirely revised for the English form as requested

Once performed these corrections, the paper can be accepted for publication.

MINOR COMMENTS

·Line 23, please specify the abbreviation “BLI” or use the specific terms in the abstract;

Authors’ reply and amendments: The abbreviation was added to the abstract as requested

·Line 127, please type “e.g.” in italic;

Authors’ reply and amendments: The term was modify as requested

·Line 152, please explain the “TKC” abbreviation.

Authors’ reply and amendments: The explanation was added as requested

Reviewer 5 Report

Comments and Suggestions for Authors

The manuscript entitled "Synergistic activity of Cefiderocol in combination with Avibactam, Sulbactam or Tazobactam against Carbapenem-Resistant Gram-Negative Bacteria"  desribed the sinergi between CFD and some 

Ther are only two minor coments for the authors.

1- World Health Organisation, just cahnged some priority microorganims, Plesae revise the comunication and adapt the manuscript.

2- The Fig 1 is dificult to read, it had very small letter in each graphic, and It is not readable

Author Response

Ther are only two minor coments for the authors.

1- World Health Organisation, just cahnged some priority microorganims, Plesae revise the comunication and adapt the manuscript.

Authors’ reply and amendments: We thank the reviewer for his work and for his comments. The list of WHO of antibiotic-resistant bacterial pathogens was modify accordingly to the 2024 WHO list, as requested

2- The Fig 1 is dificult to read, it had very small letter in each graphic, and It is not readable

Authors’ reply and amendments: The figure is uploaded separately to improve the resolution and to render it more readable

Round 2

Reviewer 3 Report

Comments and Suggestions for Authors

- Authors’ reply and amendments:Most of the strains used in this manuscript were characterized in our previous study. However, in this manuscript we tested different antimicrobial combinations and also using reference and more accurate testing method for synergy (i.e. TKC).

- This information should clearly appear in the introduction or method part. Thus the reader may refer to the previous study, and better understands why you selected the same strains.

- Authors’ reply and amendments: The discrepancies observed by the reviewer between the MICs in these studies were due to the different testing methods used. In particular, in previous study we tested all the strains by using by e-test method, while in this study we performed the microdilution method using Iron Depleted-Mueller Hinton Broth (ID-MHB)

- Please add to Table S1: Results obtained by Etest method (referring to the previous study), next to results obtained by microdilution in the present study (i.e. there will be the FDC column for microdilution, and next to it a FDC column for etest)

- Authors’ reply and amendments: The TKC experiments for antimicrobial combinations were performed in triplicate for the principal time points (0, 4 and 24 hours) to confirm the data and in single for the middle time points.

-Please add this precision in the method part (part 2.3)

- Authors’ reply and amendments: We apologize for the mistake. We modify the Table 1 accordingly to the correct antimicrobial resistance genes for each species

-The invertion is still here (between PA and ABA lines), please be careful.

- Authors’ reply and amendments: We thanks reviewer for his insightful comment. We tested the strains at 24h for cefiderocol alone and we didn’t observed any substantial difference rather than parental strains. At the same time, it is possible to hypothesize that resistant subpopulation could emerge under sublethal concentrations of cefiderocol that could related to emerging resistant clones. However, we didn’t tested this hypothesis and further study will be conducted to better evaluate this important point raised by the reviewer

-Please add this discussion concerning a potential regrowth (in Results or Discussion)

- Authors’ reply and amendments: We agree with the reviewer comment, that the TKC cannot be used in a classical routine workflow principally based on the laborious and time-consuming work for this type of experiment. Based on our experience, we could hypothesize that e-test method could be use for synergy testing in a bacteriological laboratory during the routine workflow. However, E-test method have different limitations that should be considered during the results interpretations. In our advise, further studies should be done to implement TKC assay to include this method in a bacteriological routine workflow.

-This could be added as a conclusion/perspective.

Author Response

Reviewer_3

Comments and Suggestions for Authors

1 - Authors’ reply and amendments: “Most of the strains used in this manuscript were characterized in our previous study. However, in this manuscript we tested different antimicrobial combinations and also using reference and more accurate testing method for synergy (i.e. TKC).“ 

- This information should clearly appear in the introduction or method part. Thus the reader may refer to the previous study, and better understands why you selected the same strains.

- Authors’ reply and amendments_Revision-2: We thank the reviewer for his comments. A sentence citing our previous work was added to the text as requested (see line 89 pag.2)

2 - Authors’ reply and amendments: The discrepancies observed by the reviewer between the MICs in these studies were due to the different testing methods used. In particular, in previous study we tested all the strains by using by e-test method, while in this study we performed the microdilution method using Iron Depleted-Mueller Hinton Broth (ID-MHB) 

- Please add to Table S1: Results obtained by Etest method (referring to the previous study), next to results obtained by microdilution in the present study (i.e. there will be the FDC column for microdilution, and next to it a FDC column for etest)

 - Authors’ reply and amendments_Revision-2: As requested in point 1, we cited our previous study in the text and we referred to the previous results. However, we opted to not add these data in the table for clarity of the table and data presentation. Indeed, we think that Table S1 shown several MIC and that by adding previous results (not tested in this study) could render the table unclear and not easy to understand for the readers.

3 - Authors’ reply and amendments: The TKC experiments for antimicrobial combinations were performed in triplicate for the principal time points (0, 4 and 24 hours) to confirm the data and in single for the middle time points.

-Please add this precision in the method part (part 2.3)

 - Authors’ reply and amendments_Revision-2: A sentence describing the used method was added to the text as requested (see lines 133-134 pag.3)

4 - Authors’ reply and amendments: We apologize for the mistake. We modify the Table 1 accordingly to the correct antimicrobial resistance genes for each species

-The invertion is still here (between PA and ABA lines), please be careful.

- Authors’ reply and amendments_Revision-2: The correct porin genes were inverted in table 1 as requested

5 - Authors’ reply and amendments: We thanks reviewer for his insightful comment. We tested the strains at 24h for cefiderocol alone and we didn’t observed any substantial difference rather than parental strains. At the same time, it is possible to hypothesize that resistant subpopulation could emerge under sublethal concentrations of cefiderocol that could related to emerging resistant clones. However, we didn’t tested this hypothesis and further study will be conducted to better evaluate this important point raised by the reviewer

-Please add this discussion concerning a potential regrowth (in Results or Discussion)

 - Authors’ reply and amendments_Revision-2: A sentence describing the limit of this study was added to the text in the discussion section as requested (see lines 276-281, pag. 7)

6 - Authors’ reply and amendments: We agree with the reviewer comment, that the TKC cannot be used in a classical routine workflow principally based on the laborious and time-consuming work for this type of experiment. Based on our experience, we could hypothesize that e-test method could be use for synergy testing in a bacteriological laboratory during the routine workflow. However, E-test method have different limitations that should be considered during the results interpretations. In our advise, further studies should be done to implement TKC assay to include this method in a bacteriological routine workflow. 

-This could be added as a conclusion/perspective.

- Authors’ reply and amendments_Revision-2:  A sentence describing the possible use of TKC and its limitation in a routine workflow was added to the text in the discussion section as requested (see lines 281-287, pag. 7)